# Experimental and Simulation Study of the Fracture Instability Behavior in Polypropylene Fiber-Reinforced Concrete

**DOI:** 10.3390/ma16134729

**Published:** 2023-06-30

**Authors:** Peng Cao, Liang Cao, Guoqing Chen, Feiting Shi, Changjun Zhou, Jianru Wang

**Affiliations:** 1Faculty of Architecture, Civil and Transportation Engineering, Beijing University of Technology, Beijing 100124, China; caopeng@bjut.edu.cn (P.C.); caol@emails.bjut.edu.cn (L.C.); 2School of Architecture and Engineering, Northeast Electric Power University, Jilin 132012, China; chenguoqing666@126.com; 3Civil Engineering Department, Yancheng Institute of Technology, Yancheng 224051, China; shifeiting@ycit.cn; 4School of Transportation & Logistics, Dalian University of Technology, No. 2 Linggong Street, Dalian 116023, China; 5The 41st Institute of the Fourth Research Academy of CASC, Xi’an 710025, China

**Keywords:** concrete fracture, plastic damage model, digital speckle correlation method, local deformation zone, cusp catastrophe theory

## Abstract

This study investigated the fracture characteristics of plain concrete and polypropylene fiber-reinforced concrete (PFRC) using pre-notched three-point bending beam tests with the digital speckle correlation method (DSCM). Then, the fracture instability behavior of the two types of beams was simulated in finite elements based on the plastic damage model and the cohesion model, for which the applicability was assessed. Furthermore, the stability of the Big Gang Mountain Dam made from plain concrete or PFRC subjected to the earth-quake loading was simulated with the plastic damage model. The results show that the limiting length of the non-local deformation zone can be used as an indicator of instability damage in a concrete structure. The simulation results of the plastic damage model agreed well with the local deformation in the pre-notched three-point bending beam test obtained from the DSCM. The plastic damage model was found to be capable of describing the residual strength phenomenon, which the cohesive model was not capable of. The damage evolution regions of the PFRC dam are strictly constrained in some regions without the occurrence of the local deformation band across the dam, and PFRC can dramatically reduce the failure risk under earthquake loading. The numerical solution proves that PFRC is an advisable material for avoiding failure in concrete dams.

## 1. Introduction

The meso-cracks existing in concrete structures always dramatically influence the safety and durability of the structure and reduce the loading capacity of the structure due to the quasi-brittle essence of the concrete material [1]. It is common that a crack will propagate and grow without more energy provided by the external conditions after the length of the crack has reached a critical value; at which point, the energy stored by the region outside of the fracture zone, being at least equal to the fracture resistance capacity of the other ligaments in the fracture process zone, will transfer and release to activate the instability failure. Actually, the length of meso-crack propagation is of importance for determining the instability failure of a concrete structure, according to recent four decades of research into concrete fracture behavior [2]. Unfortunately, the instability failure of the concrete structure, including building, pavement and dam structures, always occurs instantaneously without exhibiting enough information except for the crack length [3,4]. Actually, tracking the length of a crack growing in a concrete structure, especially for plain concrete, is also complicated due to the accuracy of the measuring equipment, as well as the fact that the crack length always increases irregularly without a fixed time rate. Generally, the failure disasters of a concrete structure induced by crack propagation are extremely catastrophic, especially for dam structures utilized for water storage, irrigation, and electricity generation [5]. Hitherto, the mechanism underlying the fracture behavior of the concrete has been an issue in civil and hydraulic engineering.

In order to identify concrete fracture behavior, various equipment has been fabricated for testing the fracture resistance capacity of concrete. The most sensitive fracture index of concrete mainly concerns the tensional fracture resistance capacity [6]. Ironically, the direct tension test is always involved and is inconvenient, due to the difficulties encountered in the fabrication of suitable and beneficial equipment. In practice, the indirect tension testing methods such as the three-point bending beam test, four-point bending beam test, indirect tension sphere disk test and indirect tension semi-sphere disk test are much more feasible for investigating the fracture resistance capacity of concrete [7,8,9]. The indirect tension sphere disk test is commonly utilized, especially for pavement engineering. This method is widely recommended for various specifications [10,11]. However, the fracture path employed by this method always exhibits a complicated fracture distribution pattern due to the fact that the fracture path is highly dependent on the aggregate distribution in front of the initialization crack and the geometry pattern of the specimen. To overcome the fault above-mentioned, an initial notch is adopted in the indirect tension semi-sphere disk test for firmly controlling the fracture path under the loading process. Generally, this modified method has been widely adopted for investigating the fracture characteristics for asphalt concrete. The three-point bending beam and four-point bending beam tests are very similar, and widely utilized in civil and hydraulic engineering for estimating the fracture resistance capacity of cement concrete. They both can be fabricated for testing type I, type II and mixed-type fracture patterns. Compared with the four-point bending beam, the three-point bending beam test is more convenient to fabricate and operate [12]. Furthermore, many studies have also calculated and collected the analytical solutions of the stress intensity factors for the three-point bending beam model in the various boundary conditions, greatly promoting the application and generalization of the three-point bending beam test [13].

Hillerberg et al. [14] and Petersson [15] are pioneers in using the three-point bending beam test to study concrete fracture characteristics, especially the fracture energy, as well as proposing the cohesive model for explaining the concrete fracture process under flexural loading conditions. Beygi et al. [16] modified the three-point bending beam specimens with an initial single notch, and tested the fracture energy of self-compacting concrete and pointed out that the relativity of fracture energies for the material was satisfied with the three-point bending beam test. Xu and Reinhardt [17,18] analyzed the nature of concrete fractures by adopting the three-point bending beam test and proposed the double parameter fracture model for describing the initial fracture and unstable fracture behaviors. Cao et al. [19] analyzed the fracture process of polypropylene fiber-reinforced concrete (PFRC) with the three-point bending beam test to investigate the influence of different mass contents of polypropylene fiber on the fracture energies. It was clearly observed that a residual strength exists, forming a platform in the load-crack mouth opening displacement curve after the peak value stage. Actually, in this situation, the traditional standard equation for calculating the fracture energies is not very suitable. Therefore, monitoring the finite length of the nonlocal deformation zone in front of the crack by means of digital speckle techniques is of vital importance in determining the unstable fracture phenomenon in concrete materials at this stage [20].

Generally, the three-point bending beam test is of importance in obtaining fracture information during crack initiation and propagation to understand the mechanism of the fracture phenomena [21]. The characteristic length of the concrete material controlling the nonlocal region was measured by means of the three-point bending beam test by Pijaudier and Bazant [22]. Li et al. [23] applied digital image correlation technology to investigate the formation of the fracture process zone and found that large aggregates lead to the enhancement of fracture toughness and fracture energy of dam concrete. Zhou et al. [24] conducted a comprehensive fine-scale study of the evolution of the fracture process zone based on a mesoscale model and found that the width of the fracture process zone was insensitive to the beam size, but that the length of the fracture process zone had a strong dependence on the beam size.

In summary, all the studies mentioned above not only focused on the evaluation of concrete fracture energy but also paid attention to describing the concrete fracture phenomena in the three-point bending beam tests. Many damage models and fracture models were also developed based on the three-point bending beam test results. Actually, more and more research has been undertaken to discover the mechanism underlying the local deformation of concrete materials in the last two decades [25,26]. Wells and Sluys [27] simulated the fracture behaviors of concrete materials by means of the cohesive model proposed by Hillerberg et al. [14], and the relative displacements of the crack tip are regarded as the parameters for judging the initiation and propagation of the crack surfaces. The common feature of the above-mentioned research is that a cohesive model is adopted in determining the fracture behavior. Regrettably, the cohesive model is only suitable for the fracture phenomenon when the fracture path is fixed, and then the cohesive elements are fabricated in this path for the crack to pass through. Yin et al. [28] provide a random failure path model for asphalt concrete fracture behavior without the enactment of the crack path, in which the cohesive models are embedded into the interface of each finite element, thus the crack can pass any regions in which the element interfaces exist. Intrinsically, this model just regards the element interface as the potential crack path, and the fracture model adopted is still based on the cohesive model. With the development of computational mechanics, many plastic damage models were proposed and utilized in investigating concrete failure phenomena [29]. Xu et al. [30] coupled a concrete plastic damage model with a generalized plastic model for rockfill materials to investigate the development of tensile damage in concrete slabs of a concrete-faced rockfill dam during an earthquake. Zhou et al. [31] developed a plastic damage model with the non-orthogonal flow rule consisting of the damaged part driven by plastic strain and the plastic part based on effective stress. Guan et al. [32] proposed a plastic damage model for the damage analysis of concrete in a deep cut-off wall and considered different tensile and compressive damage evolution processes based on the damage factor introduced by the Sidoroff energy equivalence principle.

The studies mentioned above have shown that plastic damage models can reasonably describe the fracture behaviors of concrete. However, few studies have described the fracture instability behaviors of plain concrete and polypropylene fiber-reinforced concrete by comparing the size of the fracture process zone, crack initiation and instability fracture between experimental and numerical results. In this paper, we investigated the pressure crack mouth opening displacement (*p*-CMOD) curve and the evolution regulation of the local deformation zone for plain concrete and PFRC in the three-point bending beam tests with a digital speckle pattern system. The numerical results with the plastic damage model were compared with the experimental data to determine the rational parameters. Subsequently, catastrophe theory was adopted for the damage evolution process to determine the instability of the concrete fracture behavior by analyzing the energy evolution regulation. Finally, concrete dams suffering earthquake loading were simulated with a plastic damage constitutive model. This research is of vital importance in determining the rationality and accuracy of the parameters as the input of the plastic damage constitutive model.

## 2. Materials and Methods

### 2.1. Fabrication of the Concrete Specimens

C25 concrete was fabricated. Crushed stone with a maximum size of 19 mm was used as coarse aggregate. River sand with a fineness modulus of 2.6 was used as fine aggregate. The cement used was 42.5 grade ordinary Portland (P.O) cement and the properties are shown in Table 1. The polypropylene fiber used is a commercial product. Polypropylene fiber mass ratios higher than 0.2% were tried but the polypropylene fiber was not easy to distribute uniformly within the PFRC after mixing [19]. Therefore, the polypropylene fiber mass ratios chosen were 0 and 0.2%, and the detailed material properties are shown in Table 2. In order to realistically reproduce the instability failure of concrete in dam projects, concrete proportions from actual dam projects were used for this experiment. The mix proportion of the concrete is listed in Table 3. The water-to-cement ratio was 0.45, and the fine-to-coarse aggregate ratio was approximately 0.67. The raw materials and gradation curves are shown in Figure 1a. An agitator and platform vibrator were employed for mixing and vibrating the concrete, respectively. The equipment versions were HJW60 (Yongmei Co., Ltd., Jinan, China) and HNT012 (Tian Teng Vibrating Machinery Co., Ltd., Xinxiang, China), as shown in Figure 1b,c, respectively. It is particularly noted that to allow for a notch of 5 mm in width and 75 mm in height to be sliced in the middle of the beam, a steel plate of the same size was fixed in the mold prematurely before the specimen was poured. Furthermore, the beams were cast with the same geometrical dimension in this study to ensure that the test results were comparable. The geometry of the three-point bending beam is shown in Figure 1d.

### 2.2. Test Method

After the specimens had been cured for 28 days, the specimens were tested for three-point bending. The three-point bending test equipment used for this test is shown in Figure 1e. This study was repeated three times for each of the two polypropylene fiber content beams to ensure the validity of the experimental results. Therefore, a total of six specimens were cast for the three-point bending tests. The rigid indenter and two supports were also employed, as shown in Figure 1e. In the experimental process, the indenter provided a vertical velocity loading, and the rolling support was also utilized for restraining the displacement of the beam in the horizontal direction, inducing type I-fracture behavior, which is the most dangerous failure pattern in a concrete material. Furthermore, the indenter and supports were brushed with Vaseline to lubricate the interface between the indenter and supports with the beam.

### 2.3. Monitoring System Utilized Digital Speckle Correlation Method

The digital speckle correlation method (DSCM) was employed to monitor the fracture behaviors of the concrete beam, including the crack mouth opening displacement, displacement and strain distribution contour, because DSCM can record the deformation of the material without interference from the equipment [33,34]. Prior to the test, a layer of white paint was applied to the surface near the notch. Subsequently, black ink was sprayed from a distance, which in turn provided an evenly distributed pattern of white-black spots on the surface of the specimen [35,36]. To achieve accurate results from the CMOD, displacement and strain distribution contour, an LED light source was applied to illuminate the surface of the beam, and two industrial video cameras with a frame rate of 16 frames every second were adopted to record the testing process. The videos were gathered and calculated with a server station to generate of the displacement field and strain field simultaneously, and the fracture process zone could be conventionally identified by means of the DSCM. It should be noted that for the investigation of the initiation and propagation of the fracture in the beam, the loading velocity was strictly controlled as 0.1 mm/min due to the brittle failure nature of the concrete material. Furthermore, this loading velocity could also guarantee that the instability of the fracture process was recorded within the limited shooting mode of the cameras. The entire testing process is shown in Figure 1e.

## 3. Plastic Damage Model for Concrete

The concrete plastic damage model proposed by Lee and Fenves [37] to account for different evolutions of stiffness under tension and compression has been commonly adopted to investigate the failure phenomenon of concrete structures [38]. The concrete plastic damage model is based on the theory of plastic flow using the yield surface proposed by Lubliner et al. [39]. The non-associated potential flow used is a Drucker-Prager hyperbolic function. The concrete plastic damage model obeys the classical theory of plasticity with the following assumptions:

Additive strain incremental decomposition:(1)$$\delta \varepsilon_{ij}^{}=\delta \varepsilon_{ij}^{e}+\delta \varepsilon_{ij}^{p}$$
where $\varepsilon_{ij}^{e}$ and $\varepsilon_{ij}^{p}$ represent elastic strain and plastic strain rates, respectively. δ represents the incremental form.

A scalar damage factor D has linked non-damage status with damage status for an incremental form of the elasticity stress-strain relation as:(2)$$\delta \sigma_{ij}^{}=(1-D)E_{ijkl}^{0}(\delta \varepsilon_{kl}^{}-\delta \varepsilon_{kl}^{p})$$
where D is defined as the scalar stiffness degradation factor; $E_{ijkl}^{0}$ is the initial stiffness of the material without damage; $\varepsilon_{kl}^{}$ and $\varepsilon_{kl}^{p}$ represent the total strain tensor and plastic strain tensor, respectively.

Yield function $F({\tilde{\sigma }}_{ij},\varepsilon_{kl}^{p})\leq 0$ obeys the Kuhn-Tucker condition:(3)$$\dot{\lambda }F=0;\dot{\lambda }\geq 0;F\leq 0$$

Non-associated potential flow rule:(4)$$\delta \varepsilon_{ij}^{p}=\delta \lambda \frac{\partial G({\tilde{\sigma }}_{ij})}{\partial {\tilde{\sigma }}_{ij}}$$
where ${\tilde{\sigma }}_{ij}$ represents the stress tensor in the effective stress space.

The model is based on two sets of uniaxial data and five additional parameters. The uniaxial data correspond to the stress-strain behavior (in compression and tension separately) after reaching yielding.

### 3.1. Effective Stress and Evolution of the Scalar Damage Factor

The effective stress tensor ${\tilde{\sigma }}_{ij}$ and the Cauchy stress $\sigma_{ij}$ are calculated by Equations (5) and (6):(5)$$\delta {\tilde{\sigma }}_{ij}^{}=E_{ijkl}^{0}(\delta \varepsilon_{kl}^{}-\delta \varepsilon_{kl}^{p})$$
(6)$$\delta \sigma_{ij}^{}=(1-D)\delta {\tilde{\sigma }}_{ij}^{}$$

For any cross-section, the coefficient (1 − D), being 0 and 1 corresponding to the total damage condition and non-damage condition, respectively, represents the loss of the effective cross-section bearing loads to the total cross-sectional area. The effective stress and Cauchy stress are equal in the non-damage condition.

Evolution regulation of the damage factor D is determined by the two factors D*_t_* and D*_c_* corresponding to tensional and compressional circumstances, respectively.

### 3.2. Yield Surface

The classical Drucker-Prager yield surface is modified by two independent parameters and two sets of data. The data sets are the uniaxial stress-strain relationship independent of tension and compression. The first parameter is the ratio of *σ_b_*_0_ biaxial compressive yield stress to *σ_c_*_0_ uniaxial yield stress, adopting the yield surface in the biaxial stress state. The second parameter *K_c_* changes the shape of the deviatoric cross-section from its circular shape into a smooth triangular shape. The following paragraph discusses the yield function used in the concrete damaged plasticity model, including an explanation of each parameter. The yield function is defined by Equation (7):(7)$$F({\bar{\sigma }}_{ij},{\tilde{\varepsilon }}^{pl})=\frac{1}{1-\alpha }(\bar{q}-3\alpha \bar{p}+\beta ({\tilde{\varepsilon }}^{pl})<{\overset{⌢}{\bar{\sigma }}}_{\max}>-\gamma <-{\overset{⌢}{\bar{\sigma }}}_{\max}>)-{\bar{\sigma }}_{c}({\tilde{\varepsilon }}_{c}^{pl})\leq 0$$
where $\bar{P}=-\frac{1}{3}\sigma_{ii}$, $\bar{q}=-\sqrt{\frac{2}{3}{\bar{S}}_{ij}{\bar{S}}_{ij}}$, $\bar{S}=\sigma_{ij}-\frac{1}{3}\sigma_{kk}$; $\beta ({\tilde{\varepsilon }}^{pl})=\frac{{\bar{\sigma }}_{c}({\tilde{\varepsilon }}_{c}^{pl})}{{\bar{\sigma }}_{t}({\tilde{\varepsilon }}_{t}^{pl})}(1-\alpha )-(1+\alpha )$. $\bar{P}$, $\bar{q}$ and $\bar{S}$ represent the effective hydrostatic pressure, the equivalent von Mises stress and the deviator of the effective stress, respectively. β is the parameter for representing the influence of the multiply axial loads on the yield surface with ${\bar{\sigma }}_{c}$ and ${\bar{\sigma }}_{t}$ as effective cohesion stress in tensional and compressional conditions, respectively. <•> represents the Macauley bracket; ${\overset{⌢}{\bar{\sigma }}}_{\max}$ is maximum eigenvalue of tensor ${\bar{\sigma }}_{ij}$; α and γ are dimensionless parameters of concrete material and function. In biaxial compression (${\overset{⌢}{\bar{\sigma }}}_{\max}\leq 0$), the yield surface reduces to the Drucker-Prager condition.

Parameter α is defined as a fraction:(8)$$\alpha =\frac{\sigma_{b0}-\sigma_{c0}}{2\sigma_{b0}-\sigma_{c0}}.$$

### 3.3. Non-Associated Potential Flow

The potential flow is a hyperbolic Drucker-Prager function influenced by 3 parameters. The first is the dilatation angle ψ, which describes the angle of inclination of the failure surface towards the hydrostatic axis measured in the meridional plane [40]. The second is ω, the eccentricity of the potential flow, which defines the rate at which the function approaches the asymptote (the linear Drucker-Prager flow potential) [41]. This eccentricity multiplied by the third parameter $\sigma_{t0}$ (the uniaxial tensile strength) corresponds to the distance (measured in hydrostatic axis) between the vertex of hyperbola and the intersection of its asymptote with the hydrostatic axis. The potential flow *G* is given by Equation (9):(9)$$G=\sqrt{{\left(\omega \sigma_{t0}\tan\psi \right)}^{2}+{\bar{q}}^{2}}-\bar{p}\tan\psi .$$

## 4. Results and Discussion

### 4.1. Nonlocal Deformation Zone Observation Result

The resolution ratio of the industrial video camera is fixed, and the shooting coverage dramatically influences the accuracy and sharpness of the result, decreasing with the increase in the coverage area. Therefore, the camera only captures the region in front of the initial crack tip because the deformation phenomena changes precisely in front of the initial crack. Figure 2 provides the evolution regulation of the strain in the horizontal direction during the entire fracture period for plain concrete. As for PFRC, the details can be found in Cao et al. [19]. In the initiation of the loading process, the strain was distributed randomly due to the heterogeneous nature of the concrete material, as shown in Figure 2a (CMOD = 0.000048 m). Undeniably, the forces applied on the concrete beam were balanced with the deformation of the beam, and this period belonged to the quasi-static process. With the loading increasing, the local deformation zone began to increase rapidly with the maximal strain of 5.6 × 10^−6^, and the reason can be attributed to the stress intensity characteristic of the material in front of the crack tip, as shown in Figure 2b (CMOD = 0.000124 m). Compared with the results in Shen et al. [42], the stress status of the material was still in the elastic deformation phase.

Compared with the stress intensity region in front of the crack tip, the strain distribution of the other region was still showing small values ranging from 1 × 10^−4^ to 1 × 10^−5^, indicating that the balance of the beam had already formed. With the loading increasing to about 3 mm, the local deformation zone began to extend dramatically and the maximal strain was about 1.7 × 10^−3^, indicating that the material in front of the crack tip was not performing in an elastic state (as shown in Figure 2c with CMOD = 0.000193 m). It also should be noted that in the *p*-CMOD curve, this stage corresponds to the end of the elastic stage and is marked on the red dotted line in Figure 2i. In Jenq and Shah [43], this phenomenon is defined as the initialization of propagation of the crack. In particular, the strain level of most regions out of the fracture process zone sustains about 1 × 10^−5^, which is much smaller than the maximal values during the loading period. This phenomenon should be ascribed to the fact that these regions are under elastic unloading status. Feng and Yu [44] claimed that when a main crack begins to propagate, the other mesocracks unload due to the balance between the deformation of the material and external force. The strain distribution contour in the horizontal direction before the formation of the macro crack is also plotted in Figure 2d (CMOD = 0.000241 m). As shown in Figure 2d, the local deformation zone was quite extended, and the strain levels in the local deformation zone reached about 0.01, being 15 times that outside of the local deformation zone, at 0.0008. These phenomena indicate that the complicated interaction interferes with the mesoperspective of the local deformation zone, which can also be regarded as the fracture process zone [45,46]. After the loading displacement reached a threshold, the macrocrack can be observed clearly in Figure 2e (CMOD = 0.00044 m), whereas the length of the local deformation zone changed little compared with the previous result, shown in Figure 2d. It can be summarized that the length of the nonlocal deformation zone extends to the limiting value to result in the instability of the fracture, and the limited length of the nonlocal deformation zone may also be regarded as an index for the indication of the instability failure of the concrete structure. At the end of the loading period, the macrocrack propagated markedly and the balance condition was lost (shown in Figure 2f (CMOD = 0.0015 m) and Figure 2g (CMOD = 0.002 m)). Finally, the strain value of most regions in the beam decayed to zero, illustrating that the beam could no longer provide resistance and fractured ultimately just shown in Figure 2h (CMOD > 0.002 m).

### 4.2. Simulation Result of the Pre-Notched Three-Point Bending Beam Test

Numerical simulations with the plastic damage constitutive model were conducted to obtain the rational parameters utilized in the model. A series of two-dimensional finite element models were fabricated and the geometry parameters were all the same with the experiment specimens except for the crack tip in which a circular chamfer was fabricated in front of crack tip to avoid a numerical convergent solution during the simulation process in Figure 3a. The indenter and supports were established with the analytic rigid body due to the extremely strong stiffness of the steel used in these components compared with the concrete material. The vertical displacement loading was applied to the indenter and the vertical displacement was restrained. Two supports were restrained in both the horizontal and vertical directions, and the friction coefficient between the support and indenter was set as 0.2 to permit relative rolling [47]. Meanwhile, the friction coefficient was also chosen as 0.2 for simulation of the relative motion between the indenter and the beam.

Nine hundred four-node plane stress reduced integral elements were adopted to establish the finite element model of the pre-notched beam, as shown in Figure 3a. The element type used was CPS4R, hence there was only one numerical integration point in each element. The type of interpolation used in finite elements is linear and the values calculated in FEM are determined in points of numerical integration. The details of the mesh in the vicinity of the notch apex are shown in Figure 3b. The regions outside of the crack tip meshed with relatively coarse elements to save the computational cost due to the elastic stress circumstances during the entire loading period. However, the relatively fine elements were performed in front of the crack tip to guarantee the accuracy of the model. The entire finite element model was meshed with the central symmetry pattern to avoid numerical error being introduced by element quality. It is well known that the mesh-dependent phenomenon results in the non-coverage problem by adopting different element sizes due to the softening behavior induced by damage evolution. In this paper, the technique from Bazant [48] was employed to partially avoid this problem through matching the element size with the fracture parameter and the element size was set as that of the nonlocal deformation zone. In addition, a viscosity coefficient of 0.0002 was used to ensure the convergence of the calculation. The elastic modulus of the concrete was defined following Equation (10) from Xu [49], and was evaluated and obtained with three-point bending beam tests.
(10)$$E=\frac{1}{tc_{i}}[3.70+32.60\tan^{2}(\frac{\pi }{2}\frac{a_{0}+h_{0}}{h+h_{0}})].$$
where $c_{i}=\frac{V_{i}}{F_{i}}$ is the initial size ratio (µm/kN), which is determined by one point from the linear section of the curve of the *p*-CMOD curve (*F* and *V* are the horizontal and vertical coordination in the *p*-CMOD curve). *a*_0_, *t*, and *h* are the length of initial crack, width and height of the beam specimens, respectively. *h*_0_ is the distance from the bottom of the beam with the displacement measurement gauge.

In the finite element model, the C25 plain concrete and C25 PFRC were both investigated with the plastic damage model and cohesion model. The value of elastic modulus for concrete, being 5 × 10^8^ Pa, was calculated by equation, and Poisson’s ratio was 0.3. It should be noted that the initial limit strengths and ultimate limit strengths, $f_{c}^{in}$ and $f_{c}^{u}$ being 17 MPa and 25 MPa, respectively, in the compression condition are necessary for the definition of the damage evolution. It assumes that damage behavior will not be generated under the compressional stress condition since the tension stress will increase adequately before the compression damage behavior exists due to Poisson’s ratio. Therefore, the damage evolution regulation for compression was not adopted in this paper. The damage evolution parameters of plain concrete and PFRC for the tension condition are defined and listed in Table 4, and the limited tension strength, $f_{t}$ is 2.8 MPa. In concrete dam engineering, the three-point beam bending test is generally applied to evaluate the fracture of concrete. The material parameters in Table 4 used in the proposed model were derived from an inverse analysis from separate characterization experiments [50]. The separate characterization experiments included a uniaxial compression test [51,52] and a uniaxial tension test [53], respectively.

Meanwhile, the cohesive model was also employed in the simulation for comparison, and the fracture energy utilized in the cohesive model was defined as 150 N/M and 200 N/M, respectively, to represent the fracture behavior of concrete with or without fiber reinforcement [54].

The results of the experimental tests for each group of beams are listed in Table 5. It can be seen that the parallel test results for both beams were similar, which indicates the validity of the test results. Here, to facilitate comparison with the simulation results, the test results that were closest to the mean values were selected for comparison with the simulated results. The comparison results are presented in Figure 3c–f, respectively.

The *p*-CMOD curves of the plain concrete beam from the experiment and simulation results of the adopted cohesive model and damage model are plotted in Figure 3c. Both numerical results agree well with the experimental results. The linear stage and the softening stage after the maximal limited loading capacity are exhibited approximately. In Figure 3d, it was obviously found that the numerical *p*-CMOD curve was slightly different from the experimental curve in the softening stage after the loading reached the maximal value. The concrete difference was in that there was a residual strength in the numerical result. It can be seen that the cohesive model exhibited a more rational simulation on the fracture behavior of plain concrete, as shown in Figure 3d.

For PFRC, the plastic damage model provided more rational numerical results (Figure 3e) compared with the cohesive model (Figure 3f) for the *p*-CMOD curve. An obvious phenomenon was that the plastic damage model exhibited a residual strength, being about 40% of the maximal strength of the material after the softening stage. For plain concrete, the fracture initiated and then extended rapidly due to the brittle nature of the material. In this process, the cohesive model reflected the cohesive stress distribution in the fracture process zone appropriately. Meanwhile, plastic deformation was not the main part in the fracture. For the PFRC, crack propagation is hindered by the fiber in the cement matrixes, as shown in Figure 3g. The response of the material was as if there was plastic deformation occurring during the fracture process. Furthermore, when the concrete lost loading capacity, the fiber in the cement matrixes still partly sustained the loading capacity. That is why there was a residual strength in the *p*-CMOD curve. Definitely, the plastic damage model describes these phenomena appropriately for the failure features of PFRC.

Based on the above discussion, it can be concluded that the plastic damage model can, to some extent, describe the residual strength of the structure after the stress peak, compared to the cohesion model. Therefore, the plastic damage model can be used to investigate the effect of residual strength on the seismic performance of large concrete structures.

### 4.3. Investigation of the Local Deformation Zone in the Numerical Simulation

For a brittle material, the occurrence of a macrocrack always indicates that the structure will fail immediately. However, the detection of crack initialization is usually complicated, as well as the evaluation of the safety of the structure. In fact, the fracture process zone appears and propagates before the macrocrack, providing a method for monitoring the failure process of brittle materials, especially concrete. Prediction of the evolution regulation of the fracture process zone contributes to an understanding of the mechanism underlying instability failure induced by a macrocrack. The cohesive model is not an advisable choice for simulating the local deformation phenomenon since the fracture process zone is modeled with a non-depth interface in the cohesive model. Oppositely, the plastic damage model can reflect the local deformation phenomenon accurately by means of the damage distribution contour. In this part, the plastic damage model is employed to simulate the local deformation zone to illustrate the failure behavior of plain concrete. In the initial loading process, the damage value in the entire beam was 0. When the loading scale reached about 27% of the ultimate loading displacement, a small damaged region, with the damage value being almost 0.9, appeared in front of the crack tip, as shown in Figure 4a. However, the length of the narrow damage band had not reached the limit length of the local deformation zone to induce an instability fracture. The tensile damage of concrete is related to the plastic strain in tension while the compressive damage of concrete is related to the plastic strain in compression. Both tensile damage and compressive damage were considered in the proposed plastic damage model. However, since the compressive strength of PFRC is three to five times that of the tensile strength, and the tensile failure is the main failure mode in the three-point bending beam test and in the earthquake resistance simulation of the concrete dam, the simulation only provides the numerical results for the tensile damage. It is true that under the three-dimensional confining pressure, compressive damage exists [53]. However, in the case of this study, the compressive damage value was found to be smaller than 0.1. With such small values to consider, no numerical results were reported for the compressive damage.

These phenomena also indicate that damage behavior usually accompanies the stress intensity status and the macrocrack is already generating in the damage evolution region. This stage corresponds to the circle spot of the *p*-CMOD curve in Figure 2i, and the spot is not the maximal loading strength, meaning that the initialization of the crack does not correspond to limited strength, which were also claimed by Shah and McGarry [55]. According to the *p*-CMOD curve, the CMOD at this spot exceeded the limit strength slightly.

The damage distribution contours of the pre-notched bending beam with different loading scales are shown in Figure 4a–d. With the loading increasing to 40% of the maximal value (Figure 4b), the damage region in front of the crack extended dramatically with the length being nearly three times that at the 27% loading scale (Figure 4a). The maximal damage value was 0.99 at this time. According to the *p*-CMOD curve in Figure 2i, the CMOD of this point had reached the limit strength already. In the experiment, the macrocrack was clear to see, as shown in Figure 2. The length of the local deformation zone reached the limited value, and the instability fracture disaster was imminent. Actually, the damage contour could not exhibit the crack, just like the extended finite element method. However, the damage value being above 0.9 is usually regarded as the macrocrack, as adopted in the phase field method [42,56], and as adopted in this paper too.

With the increase in the loading scale, the evolution regulation of the damage region can be summarized as two main features: (1) the region with the damage behavior, being above 0.9, increases in the vertical direction; (2) the other regions without the damage behavior will manifest the elastic unloading conditions. Finally, a narrow damage band, in which the damage values are all above 0.9, will cross through the whole beam forming an obvious failure surface in the beam, as shown in Figure 4d. This phenomenon can be regarded as the ultimate failure of the beam.

To judge the fracture type, the horizontal and vertical strain distribution contours at the ultimate stage of the loading process are also plotted in Figure 4e,f. It was clear that the magnitudes of the strain in front of the crack tip were 0.0004 and 0.3 in the vertical and horizontal directions, respectively. Obviously, it is impossible for the strain on the concrete to reach 0.3, at which point the concrete material has already fractured, proving that the damage model is effective for simulating and describing the concrete fracture features. Meanwhile, the magnitude of the horizontal strain was about 1000 times that in the vertical direction, indicating the fracture characteristic of three-point bending beam strictly belongs to a type I fracture.

The stress and stress tensor distribution contours of the pre-notched bending beam at the ultimate stage of the loading process are presented in Figure 4g–l. It can be seen in Figure 4g–l that the max stress and stress tensor in the pre-notched bending beam when the ultimate state was reached were distributed near the top indenter, which indicates that the concrete here was damaged by compression. However, no stress distribution contours were observed in the vicinity of the notch since the crack expansion would have released the stresses during the simulation.

### 4.4. Discussion of the Fracture Stability Based on the Investigation of the Inelastic Energy

Various kinds of energy evolution curves including damage energy, external energy, internal energy and plastic energy are plotted in Figure 5a for the plain concrete three-point bending beam. The entire curves can be divided into two main stages as the elastic stage and nonelastic stage. The demarcation point can be assumed as 30% of the total loading process, in which the damage just appears in front of the crack tip corresponding to Figure 4a. Before the demarcation point, the external energy value is always greater than that of the internal energy due to the loss of energy absorbed by the supports, and the plastic energy and damage energy are 0. In particular, the entire beam performs as a conservative system and the internal energy is stored as strain energy. After the loading scale exceeded the demarcation point, the damage energy and plastic energy appeared synchronously, and the internal energy was still increasing. By contrast, the strain energy tended to drop remarkably. These phenomena illustrate that the permanent deformations induced by the dissipation mechanism happened in the beam and some region of the beam was under the unloaded condition.

It should be emphasized that the demarcation point in various kinds of energy curves just corresponds to the step in which the damage behavior begins to generate in the numerical model. Therefore, it is exquisitely rational to believe that the appearance of the damage can also be adopted to calibrate the boundary between the elastic and nonelastic stages of the energy curves. In particular, the evolution regulation in the second stage is of vital importance in determining the instability of the beam structure; thus, it would be worth investigating more carefully. We just focused on the damage energy curve and plastic energy curve since the instability fracture is relevant to the evolution regulation of the two kinds of energies. The slopes of the curves manifest a sharp change regulation ranging from a large value to a constant value with the increase in the loading scale.

Cusp catastrophe theory was adopted to investigate the stability of the dissipation behavior occurring in the simulation for determining the instability demarcation point. Firstly, the loading scale, being 27% of the entire loading, was regarded as the initial point of the non-elastic stage (second stage). In the numerical simulation, the displacement loading type was adopted on the indenter. The assumption that the length increment of the local deformation zone is proportional to the displacement scale is proposed to obtain Equation (11).
(11)$$\frac{\partial W_{i}}{\partial a}=\frac{\partial W_{i}}{k\partial \lambda }$$
where *W* represents the nonelastic energy which can be chosen as plastic energy, *W_p_*, damage energy, *W_d_* or the sum of plastic energy and damage energy. *a* and *λ* are the length of the local deformation zone and loading scale factor, respectively. *k* is defined as $k=\frac{a}{\lambda }$. In the catastrophe theory, Equation (11) can be derived as:(12)$$\frac{\partial W_{i}}{\partial a}=\frac{\partial W_{i}}{k\partial \lambda }=f(\lambda )$$

In the Taylor expansions, Equation (12) can be written as:(13)$$\frac{\partial W_{i}}{\partial \lambda }=\sum_{i=1}^{n}c_{i}\lambda^{i}$$

Due to the regulation of catastrophe theory for determining the governing equation, only the first four terms are retained as Equation (14).
(14)$$\frac{\partial W_{i}}{\partial \lambda }=c_{0}\lambda^{0}+c_{1}\lambda^{1}+c_{2}\lambda^{2}+c_{3}\lambda^{3}$$

Through the coordinate transformation regarding y=λ−q and $q=\frac{a_{2}}{3a_{1}}$, Equation (15) can be obtained.
(15)$$\frac{\partial W_{i}}{\partial y}=b_{0}+b_{1}y^{1}+b_{3}y^{3}$$
where $b_{0}=a_{4}-a_{3}q+a_{2}q^{2}-a_{1}q^{3}$, $b_{1}=a_{3}q+2a_{2}q-3a_{1}q^{2}$ and $b_{3}=a_{1}$.

When the balance condition is satisfied, Equation (15) is equal to 0:(16)$$\frac{\partial W_{i}}{\partial y}=b_{0}+b_{1}y^{1}+b_{3}y^{3}=0$$

To judge the stability of the damage energy, a three-order polynomial is employed to match the damage energy evolution curve. When the equation loses instability, the sign of the discriminant format, Δ, will change from positive to negative or from negative to positive. The Δ is deduced as:(17)$$\Delta =4u^{3}+27v^{2}$$
where $u=\frac{b_{1}}{b_{3}}$ and $v=\frac{b_{0}}{b_{3}}$.

It is necessary to divide the nonelastic stage into two sub-stages to determine the bifurcation point of Equation (17). For this simulation result, the point with 45% loading scale of the entire loading was chosen since the slope of the damage energy curve is maintained as a constant value, indicating that a steady circumstance of the damage evolution occurs after that.

The first sub-stage and second sub-stage of the damage energy evolution curve were fitted with Equations (18) and (19):
(18)$$\frac{\partial W_{i}}{\partial \lambda }=400\times (-3.0570)\lambda^{3}+300\times 4.7546\lambda^{2}+200\times (-2.8612)\lambda^{1}+80.70$$
(19)$$\frac{\partial W_{i}}{\partial \lambda }=3.9710+2\times 0.1953\lambda^{1}-3\times 5.9499\lambda^{2}+4\times 3.7658\lambda^{3}$$

The polynomial fitting curves for the first and second sub-stages are plotted in Figure 5b,c, the correlation coefficients of which are 0.96 and 0.97, respectively. The Δ values of Equations (18) and (19) calculated based on Equation (17) are negative and positive, respectively, indicating that the instability behavior occurred between the first and second sub-stages based on the criterion from Zeeman [57,58]. From the concrete fracture theory, the maximal strength is always regarded as the loss of stability, and the judgement of instability obtained with the cusp catastrophe theory agrees well with that obtained with the fracture mechanics theory.

### 4.5. Application of PFRC with Plastic Damage Model

A finite element model for the double curvature arch concrete dam, located on the Big Gang Mountain was performed with a dynamic earthquake loading of 7.0 magnitude. The plain concrete and the PFRC were considered as the materials for the dam, separately.

The body of the dam and surrounding rock are meshed with hexahedron elements (C3D8R), and the interface model [41] was adopted for modeling the transverse joint. Due to the earthquake loading, the radiation damping of the infinite soil foundation was simulated with an artificial visco-elastic boundary model, as proposed in Pan et al. [59]. In practice, 28 transverse joints were fabricated into the dam body with each section of the dam being about 20 m. The elements of each section were fabricated with about four layers and five layers in the length and depth directions, respectively. To investigate the influence of the damage region on the stability of the dam, the element size was strictly controlled to about 2.5 m in the three directions. Some 37,150 hexahedron elements and 54,053 nodes, 162,159 elements in total, were adopted in the simulation, and the element numbers for the dam body and surrounding rock were 26,265 and 10,885, respectively, as shown in Figure 5d.

In the numerical simulation process, the C25 plain concrete and C25 PFRC were both investigated with the plastic damage model. The earthquake loading was sustained for 20 s. The damage parameters for the definition of the plain concrete and PFRC are listed in Table 4. It should be noted that a dynamic enhanced factor set at 2 was adopted to represent the dynamic strengthening effort in the earthquake loading.

The damage contours after the earthquake for the numerical tests for plain concrete and PFRC are listed in Figure 5e. For plain concrete, the maximal damage factor of the dam was 1.2, which should be at most 1.0, indicating that the damage behavior had evolved adequately. In particular, the damage regions with the damage factor being at least 0.9, were mainly found on the dam heel, which is the most important region for determining the stability of the dam structure. Furthermore, the damage distribution contour also obviously exhibited that the damage band had crossed the dam body in the middle section, becoming a serious threat for the dam break. In summary, it is rational to believe that a plain concrete dam is not safe and stable when subjected to a 7.0-magnitude earthquake loading.

For the PFRC dam, the maximal damage value increased to 1.5, indicating the damage region of the dam structure was much more vulnerable. Fortunately, there was no local deformation zone in the dam, and the region with a higher damage factor was just a corner. Meanwhile the damage region areas in the PFRC dam heel were much smaller than those in the plain concrete dam heel, as shown in Figure 5e. Additionally, these damage regions just occurred in some separate areas without forming a connected region that would threaten the safety of the dam, proving that the PFRC is advisable and desirable for avoiding instability in a dam subjected to earthquake loading.

Figure 5f shows the maximal principle strain distribution contour of the plain concrete dam and PFRC dam after the earthquake. Obviously, the stain intensity regions occur in the dam heel, similar to those in the damage distribution contour, and the maximal principle strain in the PFRC dam was smaller than the plain concrete dam.

## 5. Conclusions

The pre-notched three-point bending beam tests were performed to investigate the fracture features of plain concrete and PFRC in the laboratory and in numerical simulation. Then, the bifurcation criterion proposed from the cusp catastrophe theory was used to determine the fracture stability of concrete. Finally, a dam with plain concrete or PFRC subjected to earthquake loading was simulated with the plastic damage model to evaluate the safety and stability of the dam. The conclusions are as follows.

The test results show that the ultimate length of the nonlocal deformation zone can be regarded as an indicator of instability damage in a concrete structure.The plastic damage model and cohesive model were found to be accurate and rational for predicting the fracture features of concrete, especially for the Type I fracture feature.The plastic damage model can suitably describe the residual strength phenomenon of PFRC, which the cohesive model is not capable of. However, the cohesive model can better reflect the cohesive stress distribution in the fracture process zone of plain concrete.The DSCM results indicate that when the fracture in the beam begins to propagate unstably, the length of the local deformation zone will reach a threshold. The corresponding numerical result also exhibits a similar regulation.The cusp catastrophe theory was adopted to investigate stability with the damage energy curve and the demarcation points in which damage evolution starts to lose stability were also identified and similar to those obtained from the experiment.The damage evolution regions of the PFRC dam were strictly constrained in some regions without the occurrence of the local deformation band across the dam and the PFRC could dramatically reduce the failure risk under the earthquake loading.

In conclusion, the plastic damage model developed in this study can successfully simulate the fracture performance of PFRC. This provides a theoretical basis for the application of PFRC in dam engineering.

## Figures and Tables

**Figure 1 materials-16-04729-f001:**
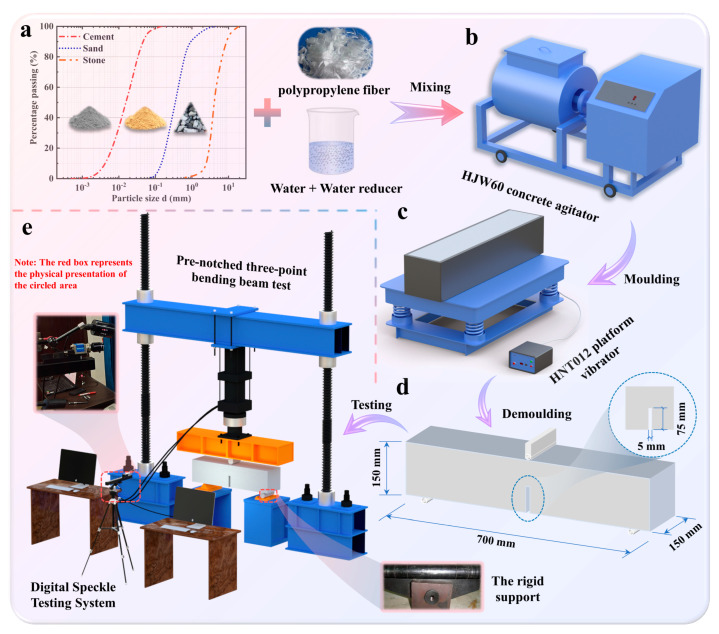
Specimen production and testing process: (**a**) raw materials and gradation curves; (**b**) concrete agitator; (**c**) platform vibrator; (**d**) specimen geometry and performance parameters; (**e**) the pre-notched three-point bending beam test monitored by the digital speckle correlation method.

**Figure 2 materials-16-04729-f002:**
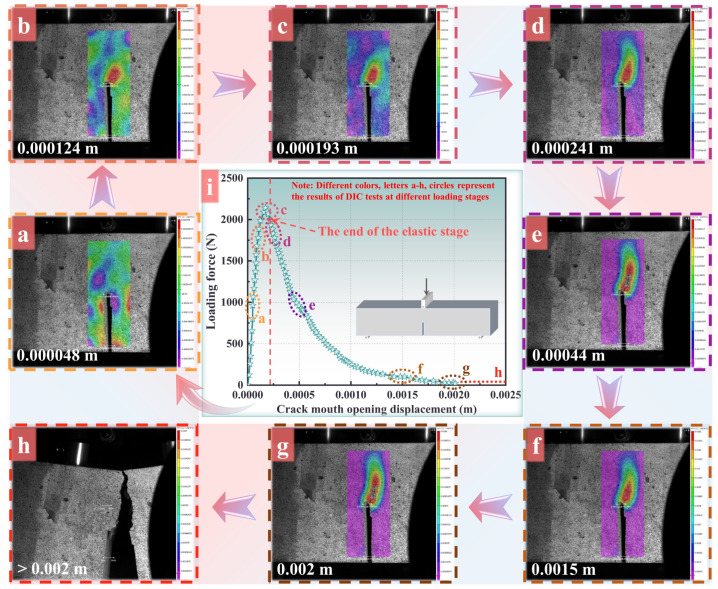
(**a**–**h**) The strain distribution contour under CMOD values in pre-notched three-point bending beam test for plain concrete; (**i**) the *p*-CMOD of the pre-notched three-point bending beam test.

**Figure 3 materials-16-04729-f003:**
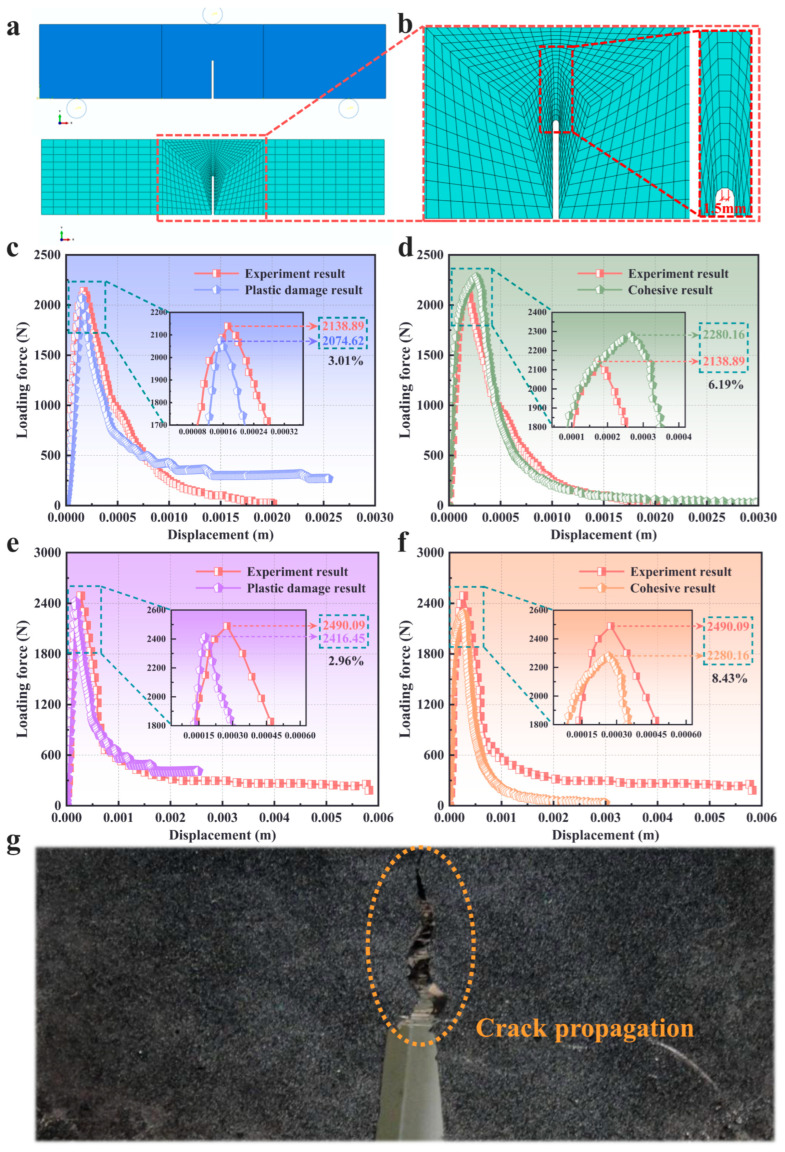
Comparison of simulation and experimental results: (**a**) the finite element model and mesh pattern of pre-notched three-point bending beam; (**b**) the details of the mesh in the vicinity of the notch apex (The red box represents a partial enlargement of the area); (**c**,**d**) numerical and experimental *p*-CMOD curves in pre-notched three point bending beams of plain concrete: (**c**) plastic damage model; (**d**) cohesive model; (**e**,**f**) numerical and experimental *p*-CMOD curves of the pre-notched three point bending beams of PFRC: (**e**) plastic damage model; (**f**) cohesive model; (**g**) the pattern of the fracture surface.

**Figure 4 materials-16-04729-f004:**
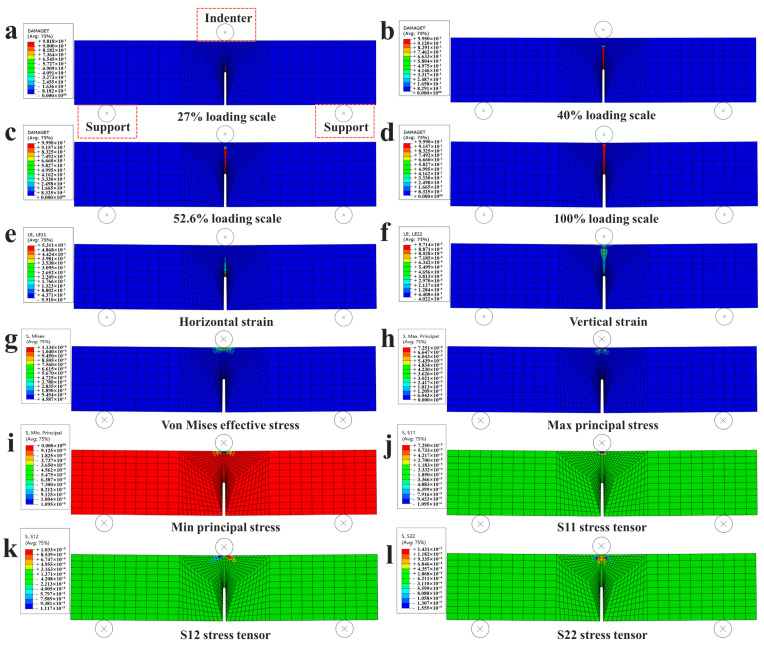
Local deformation zone results in numerical simulations: (**a**–**d**) the damage distribution contour of the pre-notched bending beam with different loading scales: (**a**) 27% loading scale; (**b**) 40% loading scale; (**c**) 52.6% loading scale; (**d**) 100% loading scale; (**e**,**f**) the strain distribution contours of the pre-notched bending beam at the ultimate stage of the loading process: (**e**) the horizontal strain distribution contour; (**f**) the vertical strain distribution contour. (**g**–**l**) The stress and stress tensor distribution contours of the pre-notched bending beam at the ultimate stage of the loading process: (**g**) the von Mises effective stress distribution contour; (**h**) the max principle stress distribution contour; (**i**) the min principle stress distribution contour; (**j**) the S11 stress tensor distribution contour; (**k**) the S12 stress tensor distribution contour; (**l**) the S22 stress tensor distribution contour.

**Figure 5 materials-16-04729-f005:**
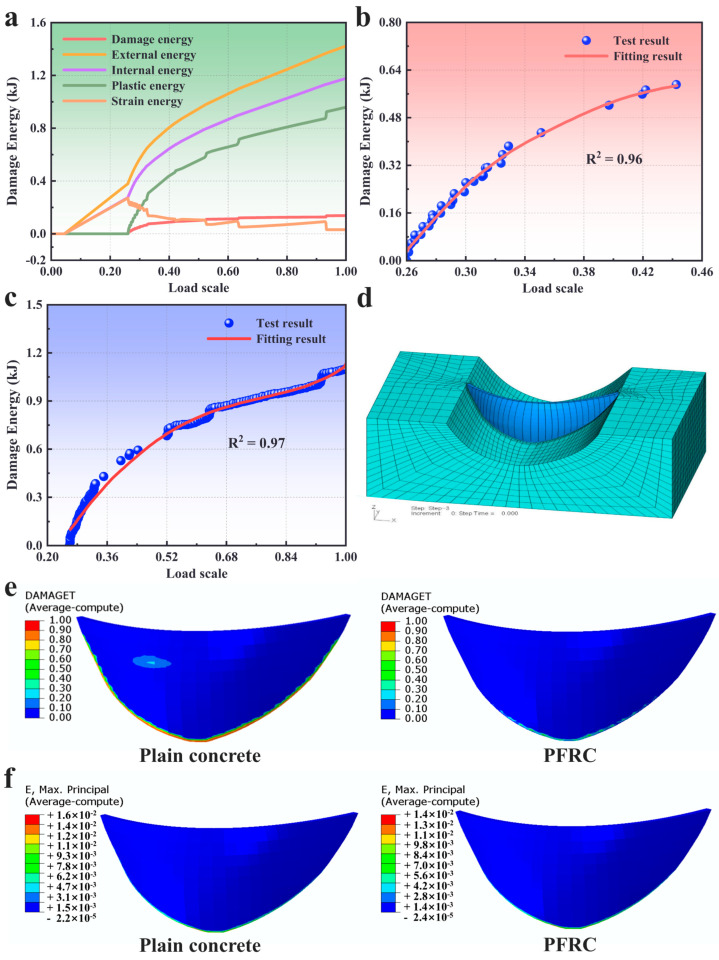
Fracture stability analysis and dam modelling results: (**a**) the evolution curves of the different energies; (**b**,**c**) the polynomial fitting curves for damage energy in the non-elastic stage: (**b**) first sub-stage; (**c**) second sub-stage; (**d**) the finite element model of the dam and surrounding rock; (**e**) the damage distribution contour of the dam; (**f**) the maximal principle strain distribution contour of the dam.

**Table 1 materials-16-04729-t001:** Properties of Portland cement.

Cement Type	Strength Grade	Rupture Strength of 28 d (MPa)	Compressive Strength of 28 d (MPa)	Soundness (mm)	Standard Consistency (%)
P.O.	42.5	7.7	45.3	1.5	27.8

**Table 2 materials-16-04729-t002:** Properties of polypropylene fiber.

Length (mm)	Diameter (mm)	Tensile Strength (MPa)	Elastic Modulus (GPa)	Specific Gravity (g/cm^3^)
12	0.048	580	3.50	0.92

**Table 3 materials-16-04729-t003:** Mix proportion of concrete (kg/m^3^).

Cement (kg)	Sand (kg)	Aggregate (kg)	Water (kg)	Water Reducer (kg)
375	768.8	1312.5	168.7	5.6

**Table 4 materials-16-04729-t004:** Damage parameters for the definition of tension damage for plain concrete and PFRC.

Concrete	Damage (-)	Inelastic Tension Strain (-)	Stress (MPa)
Plain concrete	0.00	0.0000	3.23 × 10^6^
	0.80	0.0002	0.70 × 10^6^
	0.90	0.0010	0.10 × 10^6^
	0.95	0.0060	0.05 × 10^6^
PFRC	0.0000	0.0000	1.61 × 10^6^
	0.0100	0.0005	0.81 × 10^6^
	0.3870	0.0015	0.53 × 10^6^
	0.5776	0.0026	0.34 × 10^6^
	0.6714	0.0036	0.27 × 10^6^
	0.7876	0.0046	0.22 × 10^6^
	0.8623	0.0056	0.19 × 10^6^
	0.9085	0.0065	0.17 × 10^6^
	0.9339	0.0076	0.15 × 10^6^
	0.9500	0.0085	0.12 × 10^6^
	0.9800	0.0100	0.10 × 10^6^
	0.9990	0.2000	0.01 × 10^6^

**Table 5 materials-16-04729-t005:** Maximum force for each group of test beams with corresponding CMOD.

Concrete	Maximum Force (N)	CMOD (m)	Average Force (N)
Plain concrete	2073.37	1.62 × 10^−4^	2149.84
	2138.89	1.72 × 10^−4^	
	2237.26	1.77 × 10^−4^	
PFRC	2362.76	2.43 × 10^−4^	2471.66
	2562.13	2.82 × 10^−4^	
	2490.09	2.75 × 10^−4^	

## Data Availability

The data used during the study are available from the first author and corresponding authors by request.

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
