# Peer review of "Experimental and Simulation Study of the Fracture Instability Behavior in Polypropylene Fiber-Reinforced Concrete"

_materials, 2023, doi:10.3390/ma16134729_

Round 1

Reviewer 1 Report

My comments are following:

1. Results presented in abstract not clear, authors should re-write this part as well as improve the English language. 

2. In introduction part, authors should highlight the novelty of this study. 

3. More details about used materials in this study should be provide in revised version. 

4. Authors should re-design the methodology part and make it more acurence. 

5. Please provide more details about mix design of proposed concrete. 

6. In conclusion, please highlight the applications of this type of concrete as well as the future recommendations. 

Manuscript not well written and English langauge should be improve before consider this study for publication. 

Author Response

Response to Reviewer 1 Comments

Point 1: Results presented in abstract not clear, authors should re-write this part as well as improve the English language.

Response 1: Thank you for your valuable advice. We have rewritten the abstract and had the English language revised by professional editors. Details of the changes are as follows:

"This study investigated the fracture characteristics of plain concrete and polypropylene fiber rein-forced concrete (PFRC) using pre-notched three-point bending beam tests with the digital speckle correlation method (DSCM). Then, the fracture instability behavior of the two types of beams was simulated in finite elements based on the plastic damage model and the cohesion model, for which the applicability was assessed. Furthermore, the stability of the Big Gang Mountain Dam made from plain concrete or PFRC subjected to the earth quake loading was simulated with the plastic damage model. The results show that the limiting length of the non-local deformation zone can be used as an indicator of instability damage in concrete structure. The simulation results of plastic damage model agreed well with the local deformation in the pre-notched three-point bending beam test obtained from the DSCM. The plastic damage model was found capable of describing the residual strength phenomenon, which the cohesive model was not capable of. The damage evolution regions of PFRC dam are strictly constrained in some regions without the occurrence of the local deformation band across the dam. And the PFRC can dramatically reduce the failure risk under the earthquake loading. The numerical solution proves the PFRC an advisable material for avoiding the failure of concrete dam."(Lines 13-27, page 1).

Point 2: In introduction part, authors should highlight the novelty of this study.

Response 2: Thank you for the introduction suggested. We have made changes to the introduction and highlighted the novelty of this study. Details of the changes are as follows:

"The studies mentioned above have shown that plastic damage models can describe the fracture behaviors of concrete reasonably. However, few studies have described the fracture instability behaviors of plain concrete and polypropylene fiber re-inforced concrete by the comparison of the size of the fracture process zone, crack ini-tiation and instability fracture between experimental and numerical result."(Lines 268-272, page 3).

"This research is of vital importance in determining the rationality and accuracy of the parameters as the input of the plastic damage constitutive model. "(Lines 280-282, page 3).

Point 3: More details about used materials in this study should be provide in revised version.

Response 3: We think this is an excellent suggestion. We have added information about the materials used, such as the gradation curves of the materials used in Figure 1a, to the revised version. Details of the changes are as follows:

"The C25 concrete was fabricated. Crushed stone with a maximum size of 19 mm was used as coarse aggregate. The river sand with a fineness modulus of 2.6 was used as fine aggregate. The cement used was 42.5 grade ordinary Portland (P.O) cement and the properties were shown in Table 1. The polypropylene fiber used is commercial products.The polypropylene fiber mass ratios higher than 0.2% were tried while polypropylene fiber was not easy to distribute uniformly in the PFRC after mixing [22]. Therefore, the polypropylene fiber mass ratios are chosen as 0% and 0.2%, and the detailed material properties are shown in Table 2. In order to realistically reproduce the instability failure of concrete in dam projects, concrete proportions from actual dam projects were used for this experiment. The mix propor-tion of concrete is listed in Table 3. The water to cement ratio was 0.45, and the fine to coarse aggregates ratio was approximately 0.67, respectively. The raw materials and gradation curves are shown in Figure 1a. "(Lines 285-286, page 3; Lines 408-417, page 4).

Table 1. Properties of Portland cement.

Cement

type

Strength grade

Rapture

strength of 28d (MPa)

Compressive

strength of 28d (MPa)

Soundness (mm)

Standard

consistency (%)

P.O.

42.5

7.7

45.3

1.5

27.8

Table 3. Mix proportion of concrete (kg/m3).

Cement (kg)

Sand (kg)

Aggregate (kg)

Water (kg)

Water reducer (kg)

375

768.8

1312.5

168.7

5.6

Point 4: Authors should re-design the methodology part and make it more acurence.

Response 4: Thanks for your suggestion. We have revised the test methodology section and added some test details. Details of the changes are as follows:

"It is particularly noted that to allow for a notch of 5 mm in width and 75 mm in height is sliced in middle of the beam, a steel plate of the same size was fixed in the mold prematurely before the specimen was poured. Furthermore, the beams were cast with the same geometrical dimension in this study to ensure that the test results were com-parable. The geometry of the three-point bending beam is shown in Figure. 1d." (Lines 419-424, page 1).

"After the specimens had been cured for 28 days, the specimens were tested for three-point bending. The three-point bending test equipment used for this test is shown in Figure 1e. This study was repeated three times for each of the two polypropylene fi-ber content beams to ensure the validity of the experimental results. Therefore, a total of six specimens were cast for the three-point bending tests." (Lines 429-433, page 4).

" Prior to the test, a layer of white paint was applied to the surface near the notch. Subsequently, black ink was sprayed from a distance, which in turn provided an evenly distributed pattern of white-black spots on the surface of the specimen [35,36]." (Lines 458-461, page 5).

Point 5: Please provide more details about mix design of proposed concrete.

Response 5: Thanks for your suggestion. We have provided a more detailed concrete mix design in the revised version. Details of the changes are as follows:

Table 3. Mix proportion of concrete (kg/m3).

Cement (kg)

Sand (kg)

Aggregate (kg)

Water (kg)

Water reducer (kg)

375

768.8

1312.5

168.7

5.6

Point 6: In conclusion, please highlight the applications of this type of concrete as well as the future recommendations..

Response 6: We are extremely grateful to reviewer for pointing out this problem. We have already highlighted the prospective application of polypropylene fiber reinforced concrete in dam engineering in our conclusions. Details of the changes are as follows:

"In conclusion, the plastic damage model developed in this study can successfully simulate the fracture performance of PFRC. This provides a theoretical basis for the application of PFRC in dam engineering."(Lines 993-995, page 19).

Reviewer 2 Report

The paper contains an excellent introduction. Congratulations to the authors.

I propose to reduce the abstract a bit - it should be more condensed.

The paper must include nomoenclature - a full list of abbreviations, symbols and markings. You can even add it at the end of your manuscript.

Please, clearly present the properties of the material. It is a composite, all material constants of the materials used to create the composite should be provided. Where does the data given in tables 1 and 2 come from? Please add the source to the table headings. Did the authors set it themselves?

Please show the figure of the notch and its dimensions. It is worth adding to the work a technical drawing of the specimen with the exact dimensions of the notch. It has to be at manuscript.

Please clearly indicate how many specimens the authors used in the research program in bending tests? Whether it was at least three specimens - or one - drawing conclusions based on one specimen is unreasonable. There must be at least three specimens. For these three specimens (or more), it is worth compiling graphs of force vs. displacement, force vs. CMOD, etc. It is worth showing in the tables the values of maximum forces or for selected characteristic points, the values of displacements and CMOD. If the geometry of the speciemns was the same, it is worth doing a comparative statistical analysis.

How was the material modeled for numerical calculations? Were the authors experimenting with uniaxial stretching? How many specimens were there, what were the comparative results, what material constants were determined. What material was modeled in FEM? What can the authors say about it?

In the description of the FEM model, it is necessary to write how many finite elements there were in the model, how many points of numerical integration in one element. What was the type of interpolation in the finite element? Where the values calculated in FEM were determined - in nodes or points of numerical integration. What was the size of the finite elements in relation to the characteristic dimensions of the specimens. How was the convergence of the numerical model assessed? Why did the authors use 4 node elements? We are dealing with bending issues - therefore 8 or 9 node finite elements are preferred - strongly recommended. 4-node elements do not guarantee the correctness of the results - unless the authors have comparative studies of various 4, 8 and 9-node meshes. Please show the mesh of finite elements in the vicinity of the notch tip - what was the division, density of elements, their sizes, were the nodes in the elements at the notch tip moved? Please post a figure of the mesh in the vicinity of the notch apex. Why the authors did not model half of the specimen - we are dealing with the axis of symmetry. How to relate the results of deformations and stresses (which the authors did not show - why? - please explain) to the reference values - what can these reference values be? What is the material model here - what are its parameters that can be considered reference in terms of stresses and strains? This is very important - without this the manuscript cannot be published. I am interested in effective stress distributions, stress tensor - principal stresses, average stresses, etc. Please add this to your manuscript.

Table 3 requires physical units - please add this to your paper.

Similar remarks concerning FEM are addressed to Section 4.5.

First of all, please correct the earlier part of the paper and resubmit it for review. I recommend a major revision.

Minor editing of English language required.

Author Response

Thank you very much for your efforts to improve our manuscript, which we have carefully revised in accordance with your comments, the details of which are attached.

Reviewer 3 Report

This manuscript is titled Simulation, however, I found some experimental tests have been conducted, so please modify the title, and if the experiments are not conducted by these authors, please clearly mention this matter in the manuscript.

·         Please mention in the manuscript clearly why the plastic damage model was superior to the cohesive model in the simulation.

·         The introduction is too long, please only describe the works that nearly can show the novelty of your work.

·         For this type of polymer material only linear behavior is considered in the simulation and is that in a glassy state?

·         Please describe the detail of spraying the specimens for DIC test, how the other makes sure that the size of the speckles is good enough for DIC photos to analyze?

·         The results are very interesting; however, the conclusion section has not covered all of the important points. Please modify and expand this section.

The reference papers are too old, the authors need to update it.

Author Response

(The authors gave the same response as above.)

Round 2

Reviewer 1 Report

Manuscript well revised 

English language is fine 

Reviewer 2 Report

The authors took into account all my corrections. I accept their responses to my comments.

I recommend the manuscript for publication.

Minor editing of English language required.